# Tri-explosophoric groups driven fused energetic heterocycles featuring superior energetic and safety performances outperforms HMX

Jie Li[1], Yubing Liu[1,2,3], Wenqi Ma[1], Teng Fei[1,2], Chunlin He [1,2,3,4] ✉ &
Siping Pang [1,2] ✉

The design and synthesis of novel energetic compounds with integrated properties of high density, high energy, good thermal stability and sensitivities is particularly challenging due to the inherent contradiction between energy and safety for energetic compounds. In this study, a novel structure of 4-amino-7,8-dinitropyrazolo-[5,1-*d*] [1,2,3,5]-tetrazine 2-oxide (BITE-101) is designed and synthesized in three steps. With the help of the complementary advantages of different explosophoric groups and diverse weak interactions, BITE-101 is superior to the benchmark explosive HMX in all respects, including higher density of $1.957 \, g \cdot cm^{-3}$, highest decomposition temperature of 295 °C (onset) among CHON-based high explosives to date and superior detonation velocity and pressure ($D$: 9314 $m \cdot s^{-1}$, $P$: 39.3 GPa), impact and friction sensitivities ($IS$: 18 J, $FS$: 128 N), thereby showing great potential for practical application as replacement for HMX, the most powerful military explosive in current use.

High energy density materials (HEDMs) present a class of materials that possess high densities and high detonation properties, since the existing contradiction between energy and safety, the pursuing of energetic compounds with good comprehensive properties is a desirable goal for researchers[1,2]. Energetic compounds, which are the major energy components in HEDMs, are structurally composed of backbones and explosophoric groups[3]. The explosophoric groups such as nitramine, azido, nitro, amino, N-O, nitroform etc. are the most commonly selected groups to increase the energy of energetic compounds. In terms of practical applications, a restricted process is required to examine their properties such as thermal stability, sensitivities, detonation velocity and pressure, etc., the worst performance of an energetic compound determines if it holds the application prospects[4,5]. The utilization of multi kinds of explosophoric groups to

manipulate substitution positions of energetic compounds to regulate the properties of energetic compounds is a commonly used method. Octogen (HMX) is the most powerful military explosive in current use[6,7], it has a density of $1.905 \, g \, cm^{-3}$, an onset decomposition temperature of 279 °C, detonation velocity of 9144 $m \, s^{-1}$, and detonation pressure of 39.2 GPa. During the last few decades, lots of efforts have been devoted to synthesize CHON-based high explosives[8–16]. As shown in Fig. 1, 152 neutral energetic molecules with a detonation velocity >9000 $m \, s^{-1}$ were collected and summarized. However, most of those high explosives decompose below 200 °C, which is difficult to meet the requirement for practical applications[4]. Furthermore, regards to the decomposition temperature, only 2,4,6-triamino-5-nitropyrimidine-1,3-dioxide (ICM-102)[17] ($T_d = 284$ °C) was reported to have a higher decomposition compared to that of HMX to date. The search

[1]Experimental Center of Advanced Materials, School of Materials Science & Engineering, Beijing Institute of Technology, Beijing 10081, China. [2]State Key Laboratory of Explosion Science and Technology, Beijing Institute of Technology, Beijing 10081, China. [3]Yangtze Delta Region Academy of Beijing Institute of Technology, Jiaxing 314019, China. [4]Chongqing Innovation Center, Beijing Institute of Technology, Chongqing 401120, China. ✉e-mail: chunlinhe@bit.edu.cn; pangsp@bit.edu.cn

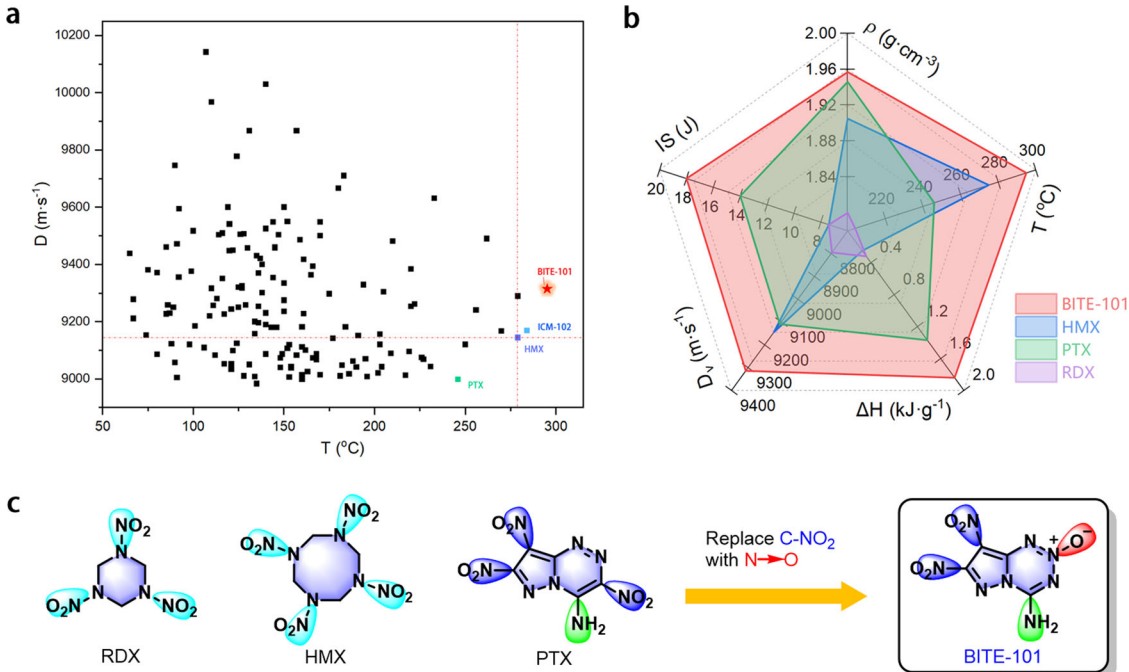

**Fig. 1 | The properties comparison of BITE-101 with other HEDMs. a** Detonation velocity vs decomposition temperature distributions chart of 152 selected neutral high explosives with detonation velocities >9000 m s$^{-1}$; **b** The Radar chart of performance comparison for BITE-101 with HMX, PTX, and RDX; **c** The structures of HEDMs.

for new energetic compounds with superior performances that outperforms HMX remains a long-term challenge.

The construction of fused ring structures is an effective way to enhance the thermal stability of energetic compounds[16,18,19]. The incorporation of a nitro group or N-O group with an amino group to create novel energetic compounds with high density, excellent thermal stability as well as low sensitivity is a well-recognized strategy[19–26]. A representative fused energetic compound 3,7,8-trinitropyrazolo[5,1-c] [1,2,4] triazin-4-amine (PTX)[24], which contains pyrazole-triazine fused ring and two kinds of explosophoric groups exhibit a high density of 1.946 g cm$^{-3}$, the high decomposition temperature of 246 °C and comparable detonation properties to that of RDX[7]. Furthermore, the existing explosophoric groups of amino and nitro in PTX offers more possibility form hydrogen bonding interaction, therefore, giving better impact and friction sensitivity (*IS*: 14 J, *FS*: 324 N) than those of RDX (*IS*: 7.5 J, *FS*: 120 N) in which the nitramine is only explosophoric group in the structure. Maintaining the diversity of explosophoric groups in backbones is beneficial to achieve balanced comprehensive properties for energetic compounds.

Our continuing goal is to seek new energetic fused ring compounds with an ideal combination of high density, detonation properties, and acceptable sensitivity toward stimuli. We proposed an enhanced explosophoric group's cooperative strategy by replacing one C-NO$_2$ of PTX with an N-O moiety, we envisioned that three different explosophoric groups could lead to diverse weak interactions while maintaining the zero-oxygen balance (CO as the product) and increasing the varieties of explosophoric group and also the heat of formation compared to that of PTX, hence make the newly designed compound with better detonation performances and higher stabilities than PTX. Herein, we present the synthesis of a novel fused ring energetic compound 4-amino-7,8-dinitropyrazolo-[5,1-*d*] [1,2,5]-tetrazine 2-oxide (named BITE−101), it was readily prepared by a three-step reaction. The structures and properties of prepared compounds were characterized by multinuclear NMR, infrared spectroscopy, elemental analysis, and differential scanning calorimetry (DSC). As expected, the more nitrogen atoms in the fused backbone and more

kinds of explosophoric groups compared to those in PTX, affording BITE-101 a high density of 1.957 g cm$^{-3}$, highest decomposition temperature (onset, 295 °C) among those CHON-based explosives with *D* > 9000 m s$^{-1}$ (Fig. 1a). The comprehensive properties of BITE-101 are significantly better than those of the other three typical energetic materials (as shown in Fig. 1b). All of the above-mentioned features with the predicted detonation velocity of 9316 m s$^{-1}$ and detonation pressure of 39.3 GPa demonstrating great potential for the application as a high energy density material.

## Results

### Materials synthesis

The synthesis of BITE-101 was commenced with the preparation of intermediate **1** by reacting 1-methyl-2-nitro-1-nitrosoguanidine[27] and 4-nitro-1*H*-pyrazole-3,5-diamine[28] in ethanol for 2 h under reflux conditions. The amino group on the 3-position of the pyrazole ring in **1** was selected and converted into a nitro group by using a mixture of 30% H$_2$O$_2$ and concentrated H$_2$SO$_4$, giving compound **2** with a yield of 45%. The treatment of **2** with 100% HNO$_3$ resulting a cyclization reaction to the formation of fused 4-amino-7,8-dinitropyrazolo[5,1-d][1,2,3,5]-tetrazine 2-oxide (BITE-101) with a yield of 86% (Fig. 2). By comparing with other methods to construct [5,6]-fused 1,2,3,5-tetrazine-2-oxide backbones which always involves the generation of highly sensitive diazo intermediates[24,25,29] or azides[22] or cyanide and other highly toxic substances[20], and also an additional *N*-oxidation step is required to access *N*-oxide[17,23]. Our new strategy to construct 1,2,3,5-tetrazine-2-oxide developed in this work is much safer and easier to be handled. The structure of BITE-101 and its intermediates were fully characterized by IR, $^1$H and $^{13}$C NMR spectroscopy as well as by elemental analysis and the spectra were listed in Supplementary Information.

### Single-crystal structure

The structure of BITE-101 was further determined by an X-ray single-crystal diffraction study of the crystals grown from methanol solution at room temperature. As shown in Fig. 3, BITE-101 crystallizes in the orthorhombic P2$_1$2$_1$2$_1$ space group, with four molecules per unit cell

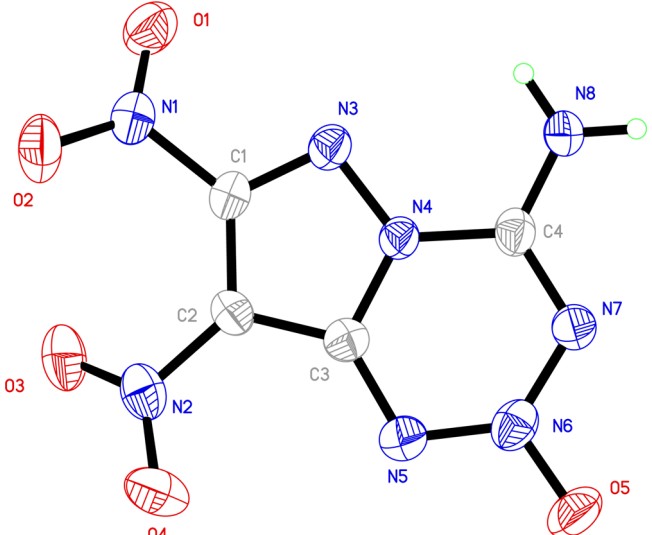

**Fig. 2 | Synthetic route for BITE-101.** BITE-101 was synthesized in three steps by starting from DANP and MNNG.

**Fig. 3 | The crystal structure of BITE-101.** The thermal ellipsoids are presented with 50% probability.

($Z$ = 4) and unit cell size V = 828.91(8) Å with a crystal density of 1.957 g cm$^{-3}$ (298 K). The C-N bond length in the pyrazole fragment lies between 1.435–1.467 Å, which is similar to those in PTX (1.436–1.458 Å)[24].

While in the 1, 2, 3, 5-tetrazine-2-oxide motif, the C-N bond length in C-NH$_2$ is 1.308 Å, the N-N bond length ranges from 1.328 to 1.356 Å, and the N-O bond length (1.240 Å) is shorter than N-O bond length in ICM-102[17] (1.253(5) Å) and PTX (1.274(8) Å). In the packing diagram of BITE-101, four independent molecules in different orientations form a wave-like stacking structure, the interlayer spacing between adjacent parallel molecules of BITE-101 is 3.165 Å, which is shorter than that in PTX (3.450 Å) (Fig. 4c, d). The two nitro groups in BITE-101 forms a torsion angle of 14.71°, smaller than that in PTX (15.68°), the shorter average hydrogen bond length was also observed (BITE-101: 2.39 Å, PTX: 2.414 Å; Fig. 4a, b), all of above-resulting BITE-101 a higher density of 1.957 g cm$^{-3}$.

## Thermostability

BITE-101 decomposes at 295 °C (onset) with a peak temperature at 300 °C (Supplementary Fig. 11), which is the highest among those reported CHON-based energetic compounds with detonation velocity >9000 m s$^{-1}$ to date. To achieve a deeper understanding of the superior thermal stability of BITE-101, the Independent

Gradient Model based on Hirshfeld partition (IGMH)[30] and bond dissociation energy (BDE) calculations were performed. As shown in Fig. 5b, each molecule in BITE-101 form one intramolecular hydrogen bond and six inter-molecule hydrogen bonds with other molecules. Based on the IGMH analysis, the green isosurface between the central molecule and surrounding molecules fragment, the larger the area between atom pairs, the stronger the interaction, indicating strong π−π and p−π interactions existing in the structure. The bond dissociation energy (BDE) calculated using the Muitifwn software through the wave function information of the molecules[31] for BITE-101 was calculated by comparing to those of PTX was shown in Fig. 5a (the detailed calculations were listed in Supplementary Table 6). The BDE value of the weakest bond in BITE-101 (C1-NO$_2$: 249.423 kJ mol$^{-1}$) is higher than that in PTX (C5-NO$_2$: 247.385 kJ mol$^{-1}$), demonstrating better thermal stability for BITE-101, which agrees well with the experimental results.

## Physicochemical properties

The heat of formation (HOF) of BITE-101 was calculated by using Gaussian 09 program[32]. The detailed calculations were provided in the Supplementary Information, and the result was listed in Table 1. Based on the measured density and calculated heat of formation for BITE-101, detonation velocity and pressure were predicted by using EXPLO5 v6.01[33]. Interestingly, BITE-101 gains a zero-oxygen balance which is the same as that of PTX and HMX when CO was used as the decomposition product. Though one less nitro group in the structure of BITE-101, possesses a much higher detonation velocity of 9316 m s$^{-1}$ than that of PTX, the detonation performances of BITE-101 is also superior to those of HMX. Furthermore, the impact sensitivity of BITE-101 (*IS* = 18 J) is significantly better than that of HMX (*IS* = 7.4 J).

## Discussion

In summary, a high energy density compound BITE-101 was synthesized through a novel cyclization strategy in three steps without involving diazo, azide highly sensitive, or other toxic intermediates. The tri- explosophoric groups (C-NO$_2$, C-NH$_2$, N-O) in the structure of BITE-101 help to form extensive hydrogen bonding and π-π interactions and result in BITE-101 featuring a high measured density of 1.957 g cm$^{-3}$, the highest decomposition temperature of 295 °C (onset) among CHON-based high explosives (*D* > 9000 m s$^{-1}$) to date, better impact sensitivity (18 J) and friction sensitivity (128 N) along with a superior detonation velocity (9316 m s$^{-1}$) to the benchmark explosive HMX, demonstrating great application prospects as a new generation of high energy density materials. The investigation of the application for BITE-101 is currently in progress in our lab.

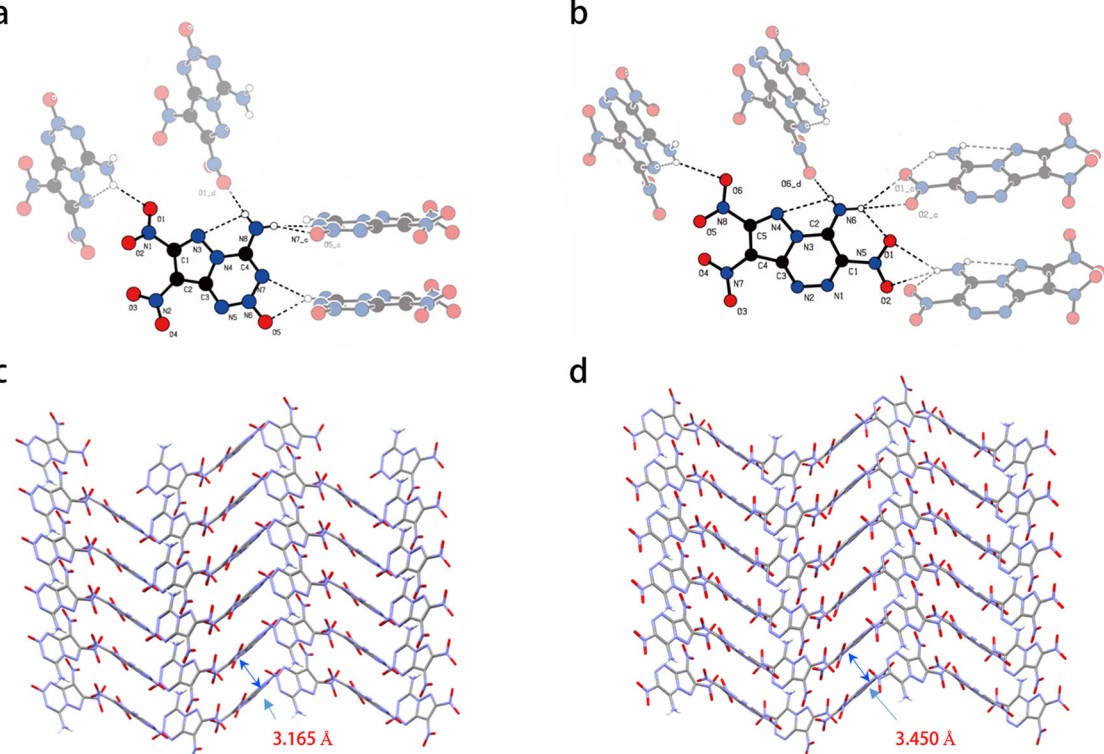

**Fig. 4 | The intermolecular H-bonds of crystals of BITE−101 and PTX with their crystal packing. a** The intermolecular H-bonds of BITE-101; **b** The intermolecular H-bonds of PTX; **c** The crystal stacking structure of BITE-101; **d** The crystal stacking structure of PTX.

## Methods

Caution! The prepared compounds are energetic materials and may explode under certain conditions. Appropriate safety precautions should be taken when preparing and handling.

### General

All reagents were purchased from Energy Chemical or Aladdin in analytical grade. DANP was prepared from the literature's method[28]. $^1H$ and $^{13}C$ NMR spectra were recorded on Bruker AVANCE 400 nuclear magnetic resonance spectrometer. DMSO-$d_6$ was employed as a solvent and locking solvent. Infrared (IR) spectra were recorded on an FT-IR spectrometer (Thermo Nicolet AVATAR 370). Decomposition (onset) temperature were recorded on a differential scanning calorimeter (DSC, TA Instruments discovery DSC25) at a scan rate of 5 °C min$^{-1}$. Elemental analyses (C, H, N) were performed on a Vario Micro cube Elementar Analyzer. Impact and friction sensitivity measurements were made using a standard BAM Fallhammer and a BAM friction tester.

### Synthesis of 3, 5-diamino-N′,4-dinitro-1H-pyrazole-1-carboximidamide (1)

1-Methyl-2-nitro-1-nitrosoguanidine (MNNG, 2 g, 0.15 mol) was added to a solution of 3,5-diamino-4-nitro-pyrazole (DANP, 2 g, 0.20 mol) in 30 ml of ethanol, and the mixture was heated to 80 °C. After refluxing for 12 h, TLC (Solvent combination: EA:PE = 1:1; $R_f$ = 0.55) detects that the reaction is completed. The precipitate was filtered off, washed with water, and dried in the air. The crude brownish-yellow solid of compound **1** was obtained (1.8 g, 55.9%). $T_d$ (onset) 237 °C. $^1H$ NMR (400 MHz, DMSO-$d_6$) δ 9.09 (d, J = 159.5 Hz, 2H), 7.55 (s, 2H), 5.90 (s, 2H). $^{13}C$ NMR (101 MHz, DMSO-$d_6$) δ 155.4, 150.9, 148.5, 109.1 ppm. IR (KBr) ṽ = 3477, 3553, 3234, 1646, 1611, 1568, 1519, 1490, 1353, 1253, 1203, 1168, 1137, 1047, 954, 820, 798, 770, 690, 647, 612,

566, 543, 409 cm$^{-1}$. Elemental analysis (%) for $C_4H_6N_8O_4$ (230.05) Calcd: C 20.88, H 2.63, N 48.69. Found: C 20.83, H 2.39, N 49.02.

### Synthesis of 5-amino-3, 4-trinitro-1H-pyrazole-1-carboximidamide (2)

About 10 mL of 30% $H_2O_2$ was added into a 100 ml flask and cooled by an ice-salt bath, 10 mL of 60% $H_2SO_4$ was added dropwise while maintaining the temperature <0 °C. Then 3, 5-diamino-N′, 4-dinitro-1H-pyrazole-1-carboximidamide (**1**, 1.15 g, 5 mmol) was added to the mixed solution at 0 °C in batches after the addition was completed. The reaction temperature gradually rose to room temperature and reacted for another 3 h. After the reaction was completed, which was monitored by TLC (Solvent combination: EA:PE = 1:2; $R_f$ = 0.40), the reaction solution was poured into ice water, and a yellow solid is precipitated. The precipitate was collected by filtration and washed with water to give **2** (0.58 g, 34.6%). $T_d$ (onset) 204 °C. $^1H$ NMR (400 MHz, DMSO-$d_6$) δ 9.88 (s, 1H), 8.82 (s, 1H). $^{13}C$ NMR (101 MHz, DMSO-$d_6$) δ 155.1, 150.2, 148.5, 109.0 ppm. IR (KBr) ṽ =3439, 3374, 3304, 3248, 1655, 1565, 1530, 1504, 1451, 1434, 1377, 1346, 1313, 1255, 1214, 1172, 1085, 1034, 948, 879, 820, 783, 773, 762, 756, 715, 683, 674, 624, 517, 462 cm$^{-1}$. Elemental analysis (%) for $C_4H_4N_8O_6$ (260.03) Calcd: C 18.47, H 1.55, N 43.08. Found: C 18.16, H 1.89, N 42.71.

### Synthesis of 4-amino-7,8-dinitropyrazolo[5,1-d][1,2,3,5]tetrazine 2-oxide (BITE-101)

To a 50 mL flask containing 3 ml 100% HNO₃ cooled to 0 °C was added 100 mg Compound **2** (100 mg, 0.384 mmol) in batches, after then, the reaction mixture was slowly warmed up to room temperature and stirred for another 3 h. After the reaction was completed it was monitored by TLC (Solvent combination: EA:PE = 1:1; $R_f$ = 0.5). Then, the reaction mixture was poured into ice water to precipitate a yellow solid. The solid was collected by filtration and washed with water to

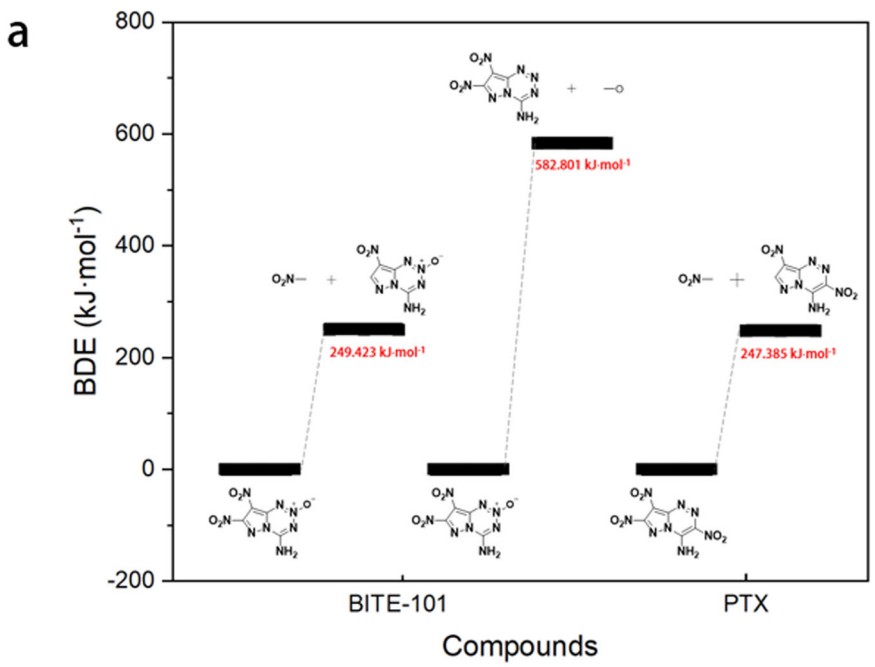

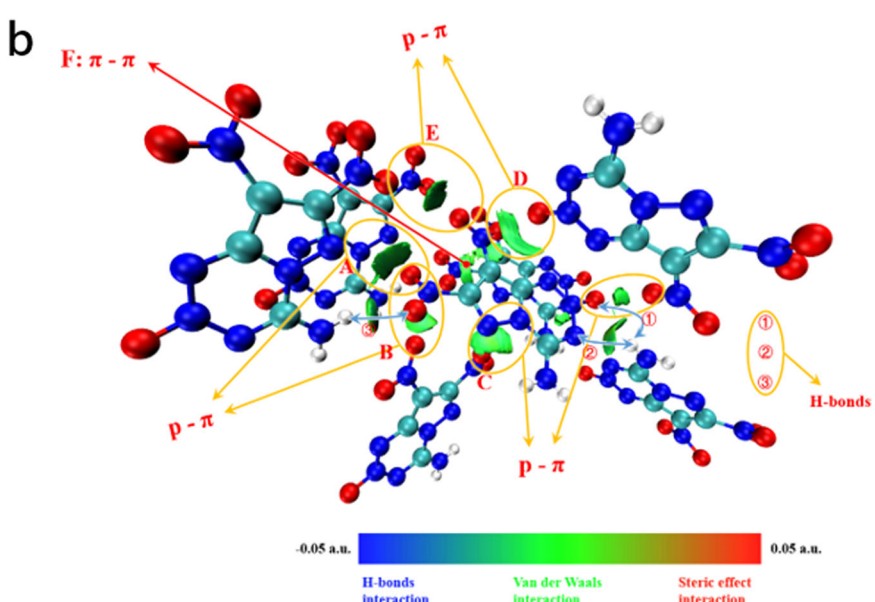

**Fig. 5 | The BDE and weak interactions analyses for BITE-101. a** The BDE of BITE-101 and PTX; **b** Intermolecular interactions analyses in BITE-101.

## Table 1 | Physicochemical Properties of BITE-101

| Comp. | $T_d{}^a$(°C) | $\rho^b$(g cm$^{-3}$) | $D_v{}^c$(m s$^{-1}$) | $P^d$(GPa) | $\Delta_f H^e$(kJ g$^{-1}$) | OB$^f$(%) | N(%) | IS$^g$(J) | FS$^h$(N) |
|---|---|---|---|---|---|---|---|---|---|
| BITE-101 | 295 | 1.957 | 9317 | 39.3 | 1.84 | 0 | 46.28 | 18 | 128 |
| PTX$^i$ | 246 | 1.946 | 9109 | 36.0 | 1.37 | 0 | 41.48 | 14 | 324 |
| HMX$^j$ | 279 | 1.905 | 9144 | 39.2 | 0.25 | 0 | 37.84 | 7.4 | 120 |

$^a$Decomposition temperature (onset).
$^b$Crystal density at 298 K.
$^c$Detonation velocity calculated using EXPLO5 v6.01.
$^d$Detonation pressure calculated using EXPLO5 v6.01.
$^e$Calculated heat of formation.
$^f$Oxygen balance based on CO for $C_aH_bN_cO_d$: OB (%) = 1600*(d-a-b/2)/Mw.
$^g$Impact sensitivity evaluated by a standard BAM fallhammer.
$^h$Friction sensitivity was evaluated by a BAM friction tester.
$^i$Ref. 24.
$^j$Refs. 7,17.

obtain a light yellow solid powder of BITE-101 (79.9 mg, 85.9%). $T_d$ (onset) 295 °C. $^1$H NMR (400 MHz, DMSO-d$_6$) δ 10.21 (d, J = 9.1 Hz, 2H). $^{13}$C NMR (101 MHz, DMSO) δ 152.8, 151.7, 146.2, 110.2 ppm. IR (KBr) ṽ = 3344, 3247, 3194, 1660, 1594, 1561, 1507, 1472, 1417, 1390, 1376, 1359, 1333, 1262, 1233, 1136, 989, 879, 855, 812, 778, 767, 735, 719, 713, 660, 643, 608, 580, 549, 460 cm$^{-1}$. Elemental analysis (%) for $C_4H_2N_8O_5$ (242.01) Calcd: C 19.84, H 0.83, N 46.28. Found: C 19.43, H 1.27, N 46.64.

## Data availability

All relevant data were included within this article and Supplementary Information. The X-ray crystallographic coordinates for BITE−101 reported in this study have been deposited at the Cambridge Crystallographic Data Centre (CCDC), under deposition number 2175186. These data can be obtained free of charge from The Cambridge Crystallographic Data Centre via www.ccdc.cam.ac.uk/data_request/cif. The data that support this study are available from the corresponding author upon reasonable request.

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

## Acknowledgements

This work was supported by the National Natural Science Foundation of China under grant No. 21875020 (C.H.) and No. 22235003 (S.P.).

## Author contributions

J.L. designed the molecular and carried out the synthesis, Y.L. and W.M. participated in the synthesis of some raw materials and properties characterization. J.L., T.F., and Y.L. finished the collection and analysis of the article Supplementary Information. J.L., C.H., and S.P. performed the structural analysis and prepared the manuscript. All authors discussed and commented on the manuscript.

## Competing interests

The authors declare no competing interests.
