## [Peer Review File · Nature Communications]

Tri-explosophoric groups driven fused energetic heterocycles featuring superior energetic and safety performances outperforms HMXReviewers' Comments:

Reviewer #1:

Remarks to the Author:

This manuscript describes the synthesis of a tricyclic fused heterocyclic material for use in energetic materials applications. Overall the work identifies a molecule, namely BITE-101 which displays properties which surpass the state of the art energetic material HMX in both measure and predicted properties.

The manuscript suffers in the following areas:

1. The manuscript has numerous typos and grammatical errors that must be corrected prior to consideration for publication
2. The use of the phrase "cask principle" is unwarranted as it is not a phrase commonly used in scientific publications of this nature. If the authors choose to use this phrase a clear explanation of what phrase means should be provided.
3. The authors claim that the synthesis described in the manuscript proceeds from commercially available materials. This is not an accurate statement and the statement should be removed.
4. An incorrect molecular name is provided in the experimental section for the synthesis of compound (1).
5. The authors claim energetic materials are derived from explosives attached to energetic backbones. This is not correct. Backbones of energetic materials tend to be inert or non-energetic.

Positives:

1. noteworthy result. Materials of this level of energy tend to have much lower thermal stabilities
2. The work is significant in the field
3. The work supports the conclusions presented
4. The methodology is sound and should be reproducible
5. There does not appear to be any flaws of concern in the data.

Recommendation:

Publication after revision

Reviewer #2:

Remarks to the Author:

This is clearly good work and deserves to get published in a top-class journal. I do not object to publication of this paper in Nature Communications.

Some comments for consideration:

- (i) Please give the names of ALL authors in the list of references (not just "et al.") in both the main paper and the SI.
- (ii) In the long table in the SI reporting explosives with VoD over 9000 m/s, please state that only neutral (not ionic) compounds have been included.
- (iii) Though the VoD and P_C-J values have been calculated using the EXPLO5 code, there is no reference given to the program in the list of references.
- (iv) If possible, please include ¹⁴/15N NMR data for BITE-101.
- (v) I understand that the authors prepared only ca. 2 g of the title compound which does not allow an experimental determination of the VoD. However, would a SSRT test be available?

Reviewer #3:

Remarks to the Author:

The work presented in this manuscript entitled "Tri-explosophoric groups driven fused energetic heterocycles featuring superior energetic and safety performances outperforms HMX" is of great interest to the researchers world over who are tirelessly working on development of new the high density-high energy materials for defense applications. The synthesized and well characterized material BITE-101 suggests that it has great potential for further investigations for its employability in the field. Single crystal x-ray proved the structure of the BITE-101 unambiguously. The work presented here is well referenced with appropriate literature references. In regards to manuscript preparation, there are some incomplete sentences (for example on page-2 of the manuscript, the first sentence of the second paragraph seems incomplete). Also there are long sentences which are more clumsy to read. For example, the second sentence of that same paragraph looks disjointed. I suggest where possible re-writing/editing the manuscript with smaller sentences with simpler language is useful for easy readability. The manuscript has many grammatical mistakes that needs to be corrected. In the experimental section, it is reported that the reactions are monitored by TLC. However, there is no mention of the solvent systems/combinations used for TLC. Authors must report the TLC solvent systems with R_f values. The yield of BITE-101 has been mentioned as 85.9%. However the quantity has not been reported. Authors must report the quantity of BITE-101. The authors have done a commendable job in well executing their research work. This work deserves publication in the esteemed journal "Nature Communications" and I strongly recommend for publication provided the manuscript address the above suggestions.

A point-by-point response to reviewer comments

Reviewer #1

Reviewer #1 (Remarks to the Author):

This manuscript describes the synthesis of a tricyclic fused heterocyclic material for use in energetic materials applications. Overall the work identifies a molecule, namely BITE-101 which displays properties which surpass the state of the art energetic material HMX in both measure and predicted properties.

The manuscript suffers in the following areas:

Q1: The manuscript has numerous typos and grammatical errors that must be corrected prior to consideration for publication

Response: We have carefully checked all through the manuscript to minimize the typos and grammatical errors.

Q2: The use of the phrase "cask principle" is unwarranted as it is not a phrase commonly used in scientific publications of this nature. If the authors choose to use this phrase a clear explanation of what phrase means should be provided.

Response: The sentence 'the evaluation follows the "cask principle"' has been deleted from the manuscript.

Q3: The authors claim that the synthesis described in the manuscript proceeds from commercially available materials. This is not an accurate statement and the statement should be removed.

Response: The statement of 'commercially available' has been deleted.

Q4: An incorrect molecular name is provided in the experimental section for the synthesis of compound (1).

Response: The name for compound 1 was corrected to 3,5-diamino-N,4-dinitro-1H-pyrazole-1-carboximidamide and updated in the manuscript.

Q5: The authors claim energetic materials are derived from explosives attached to energetic backbones. This is not correct. Backbones of energetic materials tend to be inert or non-energetic.

Response: Thanks for your suggestion, the word 'energetic' was deleted.

Positives:

1. noteworthy result. Materials of this level of energy tend to have much lower thermal stabilities
2. The work is significant in the field
3. The work supports the conclusions presented

4. The methodology is sound and should be reproducible
5. There does not appear to be any flaws of concern in the data.

Recommendation:

Publication after revision

Response: Thank you for your positive comments.

Reviewer #2

This is clearly good work and deserves to get published in a top-class journal. I do not object to publication of this paper in Nature Communications.

Some comments for consideration:

Q1: Please give the names of ALL authors in the list of references (not just "et al.") in both the main paper and the SI.

Response: The name of all authors in the references were added in both manuscript and SI.

Q2: In the long table in the SI reporting explosives with VoD over 9000 m/s, please state that only neutral (not ionic) compounds have been included.

Response: The state of neutral compounds was added in the Caption of **Supplementary Table 7** and **Fig. 1**.

Q3: Though the VoD and P_{C-J} values have been calculated using the EXPLO5 code, there is no reference given to the program in the list of references.

Response: The reference for EXPLO 5 code was added as reference 33.

Q4: If possible, please include ¹⁴/15N NMR data for BITE-101.

Response: Thanks for your suggestion, ¹⁴/15N NMR data for BITE-101 has been added in the SI (**Supplementary Fig.9** for ¹⁵N NMR; **Supplementary Fig.10** for ¹⁴N NMR)

Supplementary Fig. 9 ^{15}N NMR spectrum of BITE-101

Supplementary Fig.10 ^{14}N NMR spectrum of BITE-101

Q5: I understand that the authors prepared only ca. 2 g of the title compound which does not allow an experimental determination of the VoD. However, would a SSRT test be available ?

Response: Thank you very much for the suggestion, unfortunately, the apparatus for SSRT test is currently not available in our campus. The EXPLO5 code has been proved to have good accuracy to predict the detonation performances for CHON based energetic compounds and has been widely used by our colleagues in the area of energetic materials to predict the detonation performances of newly synthesized energetic compounds.

Reviewer #3

The work presented in this manuscript entitled "Tri-explosophoric groups driven fused energetic heterocycles featuring superior energetic and safety performances outperforms HMX" is of great interest to the researchers world over who are tirelessly working on development of new the high density-high energy materials for defense applications. The synthesized and well characterized material BITE-101 suggests that it has great potential for further investigations for its employability in the field. Single crystal x-ray proved the structure of the BITE-101 unambiguously. The work presented here is well referenced with appropriate literature references.

Q1: In regards to manuscript preparation, there are some incomplete sentences (for example on page-2 of the manuscript, the first sentence of the second paragraph seems incomplete). Also there are long sentences which are more clumsy to read. For example, the second sentence of that same paragraph looks disjointed. I suggest where possible re-writing/editing the manuscript with smaller sentences with simpler language is useful for easy readability. The manuscript has many grammatical mistakes that needs to be corrected.

Response: Thank you very much for your suggestion. We have carefully checked all through the manuscript to minimize the typos and grammatical errors.

Q2: In the experimental section, it is reported that the reactions are monitored by TLC. However, there is no mention of the solvent systems/combinations used for TLC. Authors must report the TLC solvent systems with Rf values.

Response: The solvent and Rf values information was added.

Q3: The yield of BITE-101 has been mentioned as 85.9%. However the quantity has not been reported. Authors must report the quantity of BITE-101.

Response: Thank you for your comments, the quantity of BITE-101 has been added.

The authors have done a commendable job in well executing their research work. This work deserves publication in the esteemed journal "Nature Communications" and I strongly recommend for publication provided the manuscript address the above suggestions.

Response: Thank you for your positive comments.